# Association between Mean Heart Rate and Recurrence Quantification Analysis of Heart Rate Variability in End-Stage Renal Disease

**DOI:** 10.3390/e22010114

**Published:** 2020-01-18

**Authors:** Martín Calderón-Juárez, Gertrudis Hortensia González-Gómez, Juan C. Echeverría, Héctor Pérez-Grovas, Claudia Lerma

**Affiliations:** 1Faculty of Medicine, Universidad Nacional Autónoma de México, Mexico City 04510, Mexico; martin.cal.j@comunidad.unam.mx; 2Department of Physics, Faculty of Sciences, Universidad Nacional Autónoma de México, Mexico City 04510, Mexico; hortecgg@ciencias.unam.mx; 3Department of Electrical Engineering, Universidad Autónoma Metropolitana, Unidad Iztapalapa, Mexico City 09340, Mexico; jcea@xanum.uam.mx; 4Department of Nephrology, Instituto Nacional de Cardiología Ignacio Chávez, Mexico City 14080, Mexico; hpgrovas@gmail.com; 5Department of Electromechanical Instrumentation, Instituto Nacional de Cardiología Ignacio Chávez, Mexico City 14080, Mexico

**Keywords:** heart rate variability, hemodialysis, recurrence plot analysis, active standing

## Abstract

Linear heart rate variability (HRV) indices are dependent on the mean heart rate, which has been demonstrated in different models (from sinoatrial cells to humans). The association between nonlinear HRV indices, including those provided by recurrence plot quantitative analysis (RQA), and the mean heart rate (or the mean cardiac period, also called meanNN) has been scarcely studied. For this purpose, we analyzed RQA indices of five minute-long HRV time series obtained in the supine position and during active standing from 30 healthy subjects and 29 end-stage renal disease (ESRD) patients (before and after hemodialysis). In the supine position, ESRD patients showed shorter meanNN (i.e., faster heart rate) and decreased variability compared to healthy subjects. The healthy subjects responded to active standing by shortening the meanNN and decreasing HRV indices to reach similar values of ESRD patients. Bivariate correlations between all RQA indices and meanNN were significant in healthy subjects and ESRD after hemodialysis and for most RQA indices in ESRD patients before hemodialysis. Multiple linear regression analyses showed that RQA indices were also dependent on the position and the ESRD condition. Then, future studies should consider the association among RQA indices, meanNN, and these other factors for a correct interpretation of HRV.

## 1. Introduction

The most common cause of mortality in patients with end-stage renal disease (ESRD) are due to cardiovascular diseases [1]. The main cardiovascular alterations in ESRD are fluid overload, high blood pressure, increased vascular resistance, all inducing the modification of the mechanisms that maintain functionality during the disease [2]. Patients with ESRD receiving hemodialysis have a chronic sympathetic hyperactivity that is necessary for blood pressure stability [3,4]. However, such augmented sympathetic drive is also related to a higher risk of mortality in ESRD [5].

The interaction among many feedback mechanisms confers the capacity of adapting to physiological stimuli, e.g., changes in position or hemodynamic challenges during hemodialysis, and coping with pathological conditions (such as ESRD) [6]. The entire set of adjustments needs well-integrated variables, different regulatory levels, and timing actions that despite appearing redundant can be absent or become modified when comparing the dynamical behavior of the system under different stimuli [6,7]. The dynamical response shown by heart rate variability (HRV) offers the possibility of inferring the acute and chronic adaptability of the cardiovascular system both in healthy subjects [8] and ESRD patients [4,9,10,11].

As the autonomic nervous system and other regulation systems modify the heart contraction and relaxation cycle, every interval between consecutive heartbeats has a different duration (a phenomenon referred to as HRV). Sympathetic hyperactivity increases mean heart rate, and modifies several linear HRV analysis indices, such as the standard deviation of the mean RR interval, increased power of the low-frequency (LF) band, decreased power of the high-frequency (HF) band, and an increased LF/HF power ratio of the spectral analysis [12].

Short-term HRV recordings are used to estimate the sympathetic and parasympathetic nervous system modulation through linear indices [12]. These linear indices are associated with clinical outcomes such as rapid chronic kidney disease progression [13], hypotension [14,15], and mortality [14,16]. Nonetheless, given that the HRV time series arises from the interactions of many physiological mechanisms, recently it has been considered that the linear methods for HRV analysis are insufficient to assess the underlying mechanisms behind such complex interactions [17,18,19]. Several nonlinear analysis methods have been developed for this aim [19]. However, most nonlinear analysis methods require long and even stationary time series. The recurrence plots quantitative analysis (RQA) is a suitable potential approach for noisy, nonstationary short-term time series [20], including HRV [21].

All linear HRV indices are dependent on the mean heart rate, as it has been demonstrated in sinoatrial cells, isolated hearts, living animals, healthy humans, and some diseases [22]. This relationship between linear HRV indices and the mean heart rate has been widely discussed over the last years [23]. In contrast, the association between nonlinear indices of HRV and the mean heart rate has been scarcely studied. As an example, the short-term fractal index (α1) has a significant correlation with the mean heart rate in healthy people [24,25]. Although such correlation is lost in ESRD patients before hemodialysis, it is recovered after hemodialysis and so considered as an indicator of cardiovascular adaptability [24,26] as well as a potential marker to differentiate between adaptive changes to achieve a precise demand response and long-term adjustments for facing pathophysiological alterations that, in other physiological contexts, could be deleterious. During an orthostatic challenge, the change from a supine to standing up position modifies some RQA indices [11]. Nonetheless, the influence of the mean heart rate on the RQA indices still has not been explored.

This work aimed to evaluate the correlation between RQA HRV indices and the mean heart rate in short-term HRV time series obtained from healthy subjects and ESRD patients confronting challenges of the cardiovascular system as introduced by active standing and also hemodialysis for the patients. We hypothesized that in addition to the effect provoked by these challenges, the mean heart rate also influences the RQA values.

## 2. Materials and Methods

### 2.1. Participants 

Twenty-nine ESRD patients were included in the study. All of them were treated with hemodialysis three times a week and were not prone to hemodynamic instability (they had less than three events of intradialysis hypotension in the previous month). Intradialysis hypotension was defined as a ≥20 mmHg decrease in the systolic blood pressure or by the manifestation of hypotension symptoms (nausea, vomiting, muscle cramps, dizziness, or fainting) associated with a ≥10 mmHg decrease in systolic blood pressure. Each hemodialysis session had a mean duration of 3.6 ± 0.5 h with a total volume removal = 3.1 ± 1.1 L. Hemodialysis vintage was 12.5 ± 10.2 months with a residual renal function of 0.9 ± 1.5 mL/min. The mean left ventricular ejection fraction was 65 ± 8% (obtained from the clinical records in the last 6 months prior to the study), laboratory results (from blood samples taken on any day when hemodialysis was not performed, within 1 month prior to the study) showed creatinine = 8.7 ± 2.5 mg/dL, potassium = 4.9 ± 0.7 mEq/L, phosphorous = 5.1 ± 1.5 mEq/dL, calcium = 8.9 ± 1.1 mg/dL, hemoglobin = 8.3 ± 2.7 g/dL, albumin = 3.9 ± 0.5 g/dL, cholesterol = 165 ± 41 mg/dL, and triglycerides = 145 ± 96 mg/dL. Patient age was 33 ± 10 years old, mean body mass index was 22.5 ± 2.7 kg/m^2^, and 12 patients were females. The ESRD etiology was systemic lupus erythematosus (n = 1), focal segmental glomerulosclerosis (n = 1), or unknown (n = 17). None of the patients had any other morbidity that could have affected the autonomic nervous system, neither supraventricular arrhythmias nor electrical conduction disorders.

Thirty healthy subjects were recruited with similar age (27 ± 8 years old, *p* = 0.07), sex (11 females, *p* = 0.35), and body mass index (24.3 ± 3.8 kg/m^2^, *p* = 0.11) demographics in comparison with ESRD patients. A routine clinical exam and electrocardiogram (ECG) recording were performed to discard any abnormalities. None of the healthy subjects were smokers, and they were not treated with any medication (including oral contraceptives in women) nor did they report any history of intense physical training.

All procedures performed followed the ethical standards of the Research and Ethics Committee of the Instituto Nacional de Cardiología Ignacio Chávez (protocol number is 02-392 and the protocol was approved on 10 July 2002) and the 1964 Helsinki declaration and its later amendments. Informed consent was obtained from all participants.

### 2.2. Hemodialysis Prescription

Hemodialysis sessions were delivered with volumetric dialysis machines (4008 H, Fresenius Medical Care, Bad Homburg, Germany) using ultrapure dialysate (HCO_3_ = 35 mmol/L, Na^+^ = 138 mmol/L, K^+^ = 2 mmol/L, Ca^2+^ = 3.5 mEq/L, and Mg^2+^ = 1.0 mEq/L) and polysulfone membranes (F-60 and F-80, Fresenius Medical Care, Walnut Creek, CA, USA). Hypertension was controlled by the strict prescription of dry body weight without using antihypertensive drugs following an approach of extracellular volume control by convection [27]. Patients were on a non-restrictive diet and did not use erythropoietin.

### 2.3. Study Protocol

Following Laborde et al. [28], to assess resting and reactivity conditions, ECG recordings from healthy subjects and ESRD patients were obtained during both baseline (supine position) and an orthostatic challenge (active standing) according to a protocol described previously [9]. A continuous ECG recording was obtained during 10 minutes in the supine position followed by a subsequent recording during a further 10 min of active standing. Following the recommendations in [12,28], the final 5 minutes of each recording were selected as data segments of supine position and active standing, respectively. Healthy subjects and ESRD patients maintained spontaneous breathing during all procedures, in accordance with [28], and blood pressure was measured by a sphygmomanometer at the end of each recording. Recordings from the same ESRD patients were obtained during baseline and active standing both before and after hemodialysis sessions. This study included comparisons between subjects (healthy subjects versus ESRD patients) and within subjects (supine position versus active standing and before hemodialysis versus after hemodialysis). All study participants were requested to refrain from any intake of coffee or alcohol during the 24 hours prior to the study.

### 2.4. ECG Signal Processing and Linear HRV Indices Estimation

The ECG recordings were digitized at 250 samples per second [12,28] with a 12-bit resolution using validated equipment [29,30]. The R wave of each heartbeat was identified by a second derivative algorithm [31] to obtain a time series of consecutive heartbeat RR intervals, also called NN or HRV, followed by a visually supervised inspection to correct artifacts and replace ectopic beats with interpolated intervals [12,30].

Linear HRV indices were calculated with ad hoc developed routines using MATLAB (version R2013a): mean cardiac period (meanNN), SDNN (standard deviation of all RR intervals), SDSD (standard deviation of successive RR differences), pNN50 (percentage of successive RR intervals with differences greater to 50 ms), LF (spectral power in the low-frequency band from 0.04 to 0.15 Hz, estimated in normalized units), HF (spectral power in the high-frequency band from 0.15 to 0.0.4 Hz, estimated in normalized units) [12]. Power spectral indices were obtained by the Fourier transform method after eliminating a linear trend, resampling at 3 Hz and applying a non-overlapped Hamming window of 300 data points (with 50% overlap).

### 2.5. Recurrence Plot-Based Indices

The main step for the visualization of recurrences in a time series or dataset is the calculation of the N × N matrix,
(1)Ri,j=Θ(εi−∥xi→−xj→∥), xi∈ℝm,i,j=1,…,N,
where N is the length of the time series, εi is a predefined threshold distance, ||·|| is a norm (e.g., the Euclidean norm), and Θ is the Heaviside function.

A multidimensional state space is reconstructed from the one-dimensional RR intervals in a time series, applying a time-delay embedding method. Each point in the reconstructed phase space represents the state of the system at a given time and is determined by the m coordinates of a given embedding dimension. To obtain all these reconstructed data, we used the tools developed by Norbert Marwan and colleagues, the Cross Recurrence Plot Toolbox for MATLAB (available from the toolbox for complex systems (TOCSY) webpage: http://tocsy.agnld.uni-potsdam.de/crp.php).

The embedding time delay for each recording was estimated as the first zero crossing of its autocorrelation function [32,33]. Figure 1 (upper panel) shows the autocorrelation function of individual recordings (thin lines) and the averaged autocorrelation of each group: healthy subjects (black thick line), ESRD before hemodialysis (green thick line), and ESRD after hemodialysis (magenta thick line). 

In the supine position, the embedding time delay was 5 for the healthy group, 9 for the ESRD patients before hemodialysis, and 10 for the ESRD patients after hemodialysis. During active standing, the subgroup embedding delay was 7 for the healthy group, 11 for ESRD patients before hemodialysis, and 12 for ESRD patients after. In addition, an embedding time delay = 8 was calculated from the averaged autocorrelation function from all recordings.

The minimum embedding dimension was estimated for each recording through the false nearest neighbors’ method (Figure 1, lower panel). In most recordings (97%), the amount of false nearest neighbors fell below 1% at the embedding dimension of 7. Therefore, we chose m = 7 as the embedding dimension for all recordings [9]. Next, the distances between individual points in the matrix corresponding to a state of the system at a given time were calculated using the option Maximum Norm−Fixed Recurrence Rate (“rr”) in the toolbox for autorecurrent plots [34]. The same threshold was selected for all recordings (<7%).

Although the visual inspection of the recurrence plot helps to recognize the different characteristics between physiological conditions, a recurrence quantitative analysis (RQA) of the generated patterns is generally preferred. Thus, we measured the following indices: determinism (the percentage of recurrence points forming diagonals from all recurrence points), mean diagonal length, longest diagonal line, entropy (Shannon entropy of the probability to find a diagonal line), laminarity (proportion of recurrence points forming vertical lines), trapping time (time in which the dynamics remains trapped in a certain state), maximal length of the vertical lines, recurrence time type 1, and recurrence time type 2 [35]. Specifically, recurrence time type 2 can detect very weak transitions with high accuracy, both in clean and noisy environments. Recurrence time type 1 has the advantage of being more robust to the noise level, and it is not sensitive to changes in the parameters of the algorithm. The recurrence types are defined based on the vertical distance between recurrence points in the recurrence plot [21]. In addition, entropy was obtained as defined by the Shannon information entropy of the line length distribution [36]. Shannon entropy is related to the amount of information needed to identify the state of the system.

The robustness of some of such chosen RQA indices has already been reported in a clinically oriented study [8]. In that work, all the assessed RQA indices (except for the longest diagonal line) behaved similarly for different recurrence rates, increasing significantly and consistently in both positions (supine and active standing) if the recurrence rate is increased. These indices are sensitive to recurrence rates in the range of 1−20% [8], and we used a fixed recurrence rate < 7% to construct the recurrence plots [9,11]. 

### 2.6. Statistical Analysis 

Categorical variables are reported as absolute values or percentages and were compared between healthy subjects and ESRD patients by either a Chi-squared test or exact Fischer’s test. Continuous variables are reported as mean ± standard deviation. These variables were transformed to a normal distribution by a natural logarithm in case of not showing normal distribution as proved by Kolmogorov–Smirnov test. Mean values were compared by t-test (two groups) or by ANOVA for repeated measures (three or more groups) followed by post-hoc test adjusted by the Bonferroni method. Pearson´s correlation coefficients were calculated between pairs of HRV indices and meanNN using all data from both positions in three groups: healthy subjects, ESRD before hemodialysis, and ESRD after hemodialysis. To assess RQA indices as dependent variables [28], and given the use of repeated observations per each subject in the calculation of these correlation coefficients, we also applied multiple stepwise regressions, thus considering the RQA indices as the outcome variables as well as the meanNN, the orthostatic position, the hemodialysis, and the ESRD group membership as the predictor variables. The orthostatic position, the analysis before or after hemodialysis, and the fact of belonging to the ESRD group (membership) were treated as categorical factors using dichotomous dummy variables. Each multiple regression model included the same sample size (N = 176). The statistical analyses were performed with the Statistical Package for the Social Sciences (SPSS) version 15.0 (SPSS Inc., Chicago, IL, USA), and *p*-value ≤ 0.05 was considered as the significant level.

## 3. Results

### 3.1. HRV Analysis with Linear Methods

Figure 2 shows the RR interval times series of a healthy subject and an ESRD patient. During the supine position, the healthy subject presents larger values of the RR interval (above 1.0 s), corresponding to a normal resting heart rate (close to 60 beats per minute). In response to active standing, the healthy subject has shorter RR intervals (indicating faster heart rate) with decreased variability. In contrast, the ESRD patient before hemodialysis has shorter RR intervals (i.e., faster heart rate) with decreased variability during the supine position. In response to active standing, the RR intervals decreased slightly (indicating a small acceleration of the mean heart rate), while the variability remained diminished. After hemodialysis, the RR intervals show that the heart rate remained accelerated with decreased variability in both positions. 

The linear HRV indices are shown in Table 1. In response to active standing, the healthy group shows faster heart rate (i.e., shorter mean NN), lower variability (smaller SDNN, SDSD, and pNN50), and increased LF with decreased HF, which lead to a predominant power of the low-frequency band: i.e., a larger Ln (LF/HF). 

Compared to healthy subjects, while in the supine position, the ESRD patients showed faster heart rates (i.e., shorter meanNN), reduced variability, and a predominant power of the low-frequency band either before or after hemodialysis. In response to active standing, ESRD patients before hemodialysis presented an increment in heart rate (i.e., the mean NN is shorter), a decrement of HF, and an increment in Ln (LF/HF). After hemodialysis, the response to active standing of ESRD patients showed increased heart rate, LF and Ln (LF/HF), and decreased pNN50. Compared to indices derived from recordings before hemodialysis, all linear indices are similar after hemodialysis, except for a faster heart rate (i.e., shorter meanNN) after hemodialysis during active standing.

### 3.2. HRV Analysis Based on the Recurrence Plot

Figure 2 shows the recurrence plots corresponding to each RR interval time series from a healthy subject and an ESRD patient. For the healthy subject, we observe the well-known reduction of the RR intervals and their variability during active standing. In the corresponding recurrence plot, there are extended global diagonal lines formed by short diagonals in a regularly spaced pattern covering the entire surface of the plot during the supine position. In contrast, during active standing, there is an increment in white zones (i.e., there are less accessible dynamic states). 

The lower panels of Figure 2 correspond to a patient before hemodialysis. There was a marked reduction in RR intervals and their variability, even in the supine position. The corresponding recurrence plot has less regularly extended diagonals with more concentration around the identity line and its parallel lines. During active standing, the recurrence plot is almost empty (i.e., there are very few accessible dynamical states). After hemodialysis (lower panels), there was a slight increment in the RR intervals when patients were in the supine position (compared to those before hemodialysis), which are also reflected in the recurrence plot. Again, during active standing, there is a reduction of the RR intervals and reachable dynamical states.

Table 2 shows the RQA indices of the RR time series obtained using the subgroup embedding time delay values described in the Methods section. For the healthy group, the following RQA indices increased in response to active standing: determinism, mean diagonal length, entropy, laminarity, trapping time, longest vertical line, and recurrence time type 2. Compared to healthy subjects, ESRD patients in the supine position and before hemodialysis show larger laminarity, trapping time, the longest vertical line, type 1 trapping time, and type 2 trapping time values. In response to active standing, ESRD patients before hemodialysis present an increment in the trapping time and the recurrence time type 2. After hemodialysis, ESRD patients exhibit a similar pattern of larger RQA indices such as that before hemodialysis. Akin to the healthy group, in response to active standing and after hemodialysis, ESRD patients show increased determinism, mean diagonal length, entropy, laminarity, trapping time, the longest vertical line, and the recurrence time type 2. Compared to recordings before hemodialysis, all RQA indices were similar after hemodialysis, except for a larger trapping time and the longest vertical length during active standing after hemodialysis.

In comparison, the estimated RQA indices, with an overall time embedding delay = 8 for all RR time series, are also shown in the appendix (Table A1). Most RQA indices indicate similar responses to active standing (i.e., most RQA indices are incremented during active standing) and present similar differences between groups (i.e., during the supine position, most RQA indices are larger in ESRD patients before hemodialysis and after hemodialysis compared to healthy subjects; during active standing, there were fewer differences between groups). 

Table A2 in Appendix A shows that RQA indices estimated with ad hoc embedding delays for each time series exhibit fewer significant changes in response to active standing (mostly in the healthy group) and fewer differences between the ESRD patients and the healthy group.

### 3.3. Correlation between meanNN and RQA Indices

The bivariate correlation analysis between meanNN and linear HRV indices is shown in Table 3. The healthy group shows a clear correlation between meanNN and time indices (positive correlation in SDNN, SDSD, and pNN50), and spectral indices LFnu (negative correlation), HFnu (positive correlation), and therefore a negative correlation between meanNN and Ln (LF/HF). The correlation between meanNN and time indices tend to smaller values in ESRD patients before hemodialysis, and the statistical significance is lost in HFnu and LF, although the correlation between Ln (LF/HF) and meanNN does reach statistical significance. The ESRD group after hemodialysis shows, similar to the healthy group, a correlation between meanNN and all linear indices.

The bivariate correlation analysis between meanNN and RQA indices is shown in Table 4. The healthy group shows a negative correlation between meanNN and all RQA indices (determinism, mean diagonal length, longest diagonal line, entropy, laminarity, trapping time, longest vertical line, and recurrence time type 2), except for the recurrence time type 1. Fewer correlations between meanNN and RQA indices are found in the ESRD group before hemodialysis (i.e., mean diagonal length, longest diagonal line, entropy, trapping time, and longest vertical line). Similar to the healthy group, ESRD patients after hemodialysis show a correlation between meanNN and all RQA indices, except for the recurrence time type 1.

The lower panels correspond to a patient before hemodialysis. There was a marked reduction in RR intervals and their variability, even in the supine position. The corresponding recurrence plot has less regularly extended diagonals with more concentration around the identity line and its parallel lines. During active standing, the recurrence plot is almost empty (i.e., there are very few accessible dynamical states). After hemodialysis (lower panels), there was a slight increment in the RR intervals during the supine position (compared to those before hemodialysis), which are also reflected in the recurrence plot. Again, during active standing, there is reduction of the RR intervals and reachable dynamical states.

The correlation between meanNN and RQA indices estimated with an overall time embedding delay = 8 for all RR time series are shown in Appendix A (Table A3). Most RQA indices show significant correlations with meanNN in all groups. Table A4 in the appendix shows that RQA indices estimated with ad hoc embedding delays for each time series exhibit fewer significant correlations with the meanNN in all groups in comparison with using the subgroup embedding time delays. For those RQA indices based on diagonals (determinism and mean diagonal length), there were no significant correlations with the meanNN in all groups.

Table 5 shows linear multiple regression analysis with the following selected HRV indices as predicted variables: one linear HRV index (SDNN) and three RQA indices estimated with subgroup embedding delays (determinism, trapping time, and recurrence time type 2). 

For each predicted variable of Table 5, the model was tested with the combined data from the healthy group and the ESRD patients (both before and after hemodialysis). Each model considered four independent variables: meanNN (as a continuous variable), active standing, ESRD condition, and hemodialysis (introduced as categorical variables). The results show that all dependent variables (SDNN, determinism, trapping time, and recurrence time type 2) are explained by meanNN, the body position, and the ESRD condition, while hemodialysis (before or after hemodialysis) had no significant contribution.

## 4. Discussion

Sympathetic activity increases the mean heart rate and modifies several linear HRV indices, such as the standard deviation of the mean RR interval, the power of the LF and HF bands, as well as the LF/HF spectral analysis power ratio [12]. In contrast, the parasympathetic activity generally provokes the opposite changes [12]. The correlation of these changes with meanNN is well established, as well as their adaptive changes before and after hemodialysis [4]. We show for the first time in this work that this correlation also exists with different nonlinear indices derived from the RQA plots. Bivariate correlations between all RQA indices and meanNN were significant in healthy subjects and ESRD after hemodialysis and for most RQA indices in ESRD patients before hemodialysis. Multiple linear regression analyses showed that, in conjunction with meanNN, RQA indices were also dependent on the position and the ESRD condition.

As we mentioned in the Introduction section, multiple interactions among feedback mechanisms confer the capacity of adapting to physiological stimuli, such as changes in position or hemodynamic challenges during hemodialysis, and to cope with pathological conditions (e.g., ESRD) [6]. The common vision is that the set of adjustments and timing actions at different regulatory levels are oriented to preserve or recover the homeostasis of an organism. However, more recently, other ideas from the study of various chronic diseases are considered. For example, there can be a variety of homoeostatic manifestations or developmental allostasis along the process of getting sick [37,38,39,40]. Some regulatory systems can become modified, or a particularly compensatory output can be lost, so when we study the dynamical behavior of the system under different stimuli, we need sensitive tools to disclose these regulatory changes and distinguish the physiological adaptability from drifting homeostatic modifications. The recurrence dynamical response of HRV (evaluated through the representation of all state variables in the phase space and the quantitative analysis of its recurrence plot) offers the possibility of inferring the acute and chronic plasticity of the cardiovascular system both in healthy subjects [8,41] and ESRD patients [9,10,11]. The physiological stress during active standing in healthy subjects increases determinism and entropy, reflecting an intensification of the interactions (or dependency) between the dynamic states. This is probably also expressed as increased autocorrelation, predominantly at low frequencies, which would also explain the increment of the recurrence time type 2, i.e., the system requires a longer time to revisit a state. Simultaneously, the RQA indices based on vertical lines show that the orthostatic stress bounds the dynamic behavior of the system within fewer phase space regions [9]. The effect of ESRD upon the cardiovascular dynamics seems to be in the same direction, acting as a stressor of the system that limits the number of phase space regions visited by the system (i.e., vertical-based indices are larger compared to those of healthy subjects even at rest in the supine position) and increasing the time required to revisit a state [9,10,11]. Given that after hemodialysis, the RQA indices show changes in the same direction of the response to active standing observed in healthy subjects, we argue that the RQA indices of ESRD patients evidence a preserved ability to adjust their cardiovascular dynamics when facing a hemodynamic challenge. In various physiological contexts, it is often considered that a balanced or fluctuating system reveals healthy conditions and when the system gets locked into a pattern or state, it is unregulated. However, we also must consider that ESRD patients with preserved cardiac function, such as those included in the present study, display a new set of physiological parameters showing a reconfiguration to a new homeostatic (or “allostatic”) condition.

The insights offered by the RQA indices regarding the cardiovascular dynamical adjustments before orthostatic stress, renal disease, or hemodialysis have been also discussed in previous work [9,10,11,41]. The present work evidences that the RQA indices are not only modified by these cardiovascular challenges, but they are also modified by the mean heart rate (i.e., meanNN). Since the meanNN is modified by many regulatory mechanisms [42,43], it can be considered as an integrative variable in itself, which becomes an important contributor to the dynamic behavior of HRV evaluated through RQA indices, as shown here. Therefore, the influence of mean heart rate (or meanNN) should be taken into consideration for the interpretation and clinical application of RQA indices, in the same way as it has been emphasized for other HRV indices [22,23,26]. Nevertheless, this influence has not been described yet in some of the current updates [19,28] of HRV guidelines. Overall, the negative correlations between RQA indices and meanNN indicate that at higher heart rates (i.e., smaller meanNN), there is an increment of RQA indices that, according to our interpretation in the previous paragraph, points toward a system that gets locked either by a reduction of the number of phase space regions visited or by an increment of the time required to revisit a state (i.e., recurrence time type 2 increases significantly as meanNN decreases).

Several approaches involving the concept of entropy have been applied to the analysis of time series (particularly for HRV time series). Methods such as maximum entropy [44], sample entropy [45], approximate entropy [46], fuzzy measure entropy [47], multiscale entropy [48], and renormalized entropy [49] have proven to differentiate pathologies and study physiological processes. Since every entropy method may have different assumptions, a systematic review with a comprehensive comparison of these methods and their application is warranted in further investigations. Moreover, the present work demonstrates the dependence of meanNN on the index of entropy estimated through the recurrence plot analysis (Shannon’s entropy). In light of this finding, it would be justified to investigate further the interpretation of the entropy estimated with other methods in relation to its dependence on the mean heart rate.

By searching the more adequate parameters to embed the HRV time series, we also found here that once employing the first zero crossing in the autocorrelation function to obtain the delay, we get remarkable information. The delay obtained for the different conditions explored seems to reveal a gradual evolution ranging from the healthy state in the supine position up to the ESRD patients in the supine position before hemodialysis and the patients in standing position after hemodialysis. Nevertheless, this aspect merits more extended research by itself.

Among the study limitations, we did not evaluate respiration to assess its influence on HRV. However, given the conditions in the study protocol (i.e., spontaneous breathing, in the supine position, or active standing without adding any activity that would represent cognitive or emotional effort), it is unlikely that the breathing rate of the studied subjects was out of the normal range, which could have affected the estimation of HRV indices (particularly within the high-frequency range) [28]. The study design did not consider repeated measures for the healthy group, which could provide the measurements in two different time points to compare with the two different time points of the ESRD patients (i.e., the recordings before and after hemodialysis). 

## 5. Conclusions

By analyzing HRV time series obtained in the supine position and during active standing from healthy subjects and ESRD patients, we show here that the mean heart rate (or meanNN) also influences the results of RQA, as it occurs for linear and other HRV indices. Future studies should then consider the association between RQA indices and meanNN as well as other factors, such as the position and ESRD condition, for a correct interpretation of HRV.

## Figures and Tables

**Figure 1 entropy-22-00114-f001:**
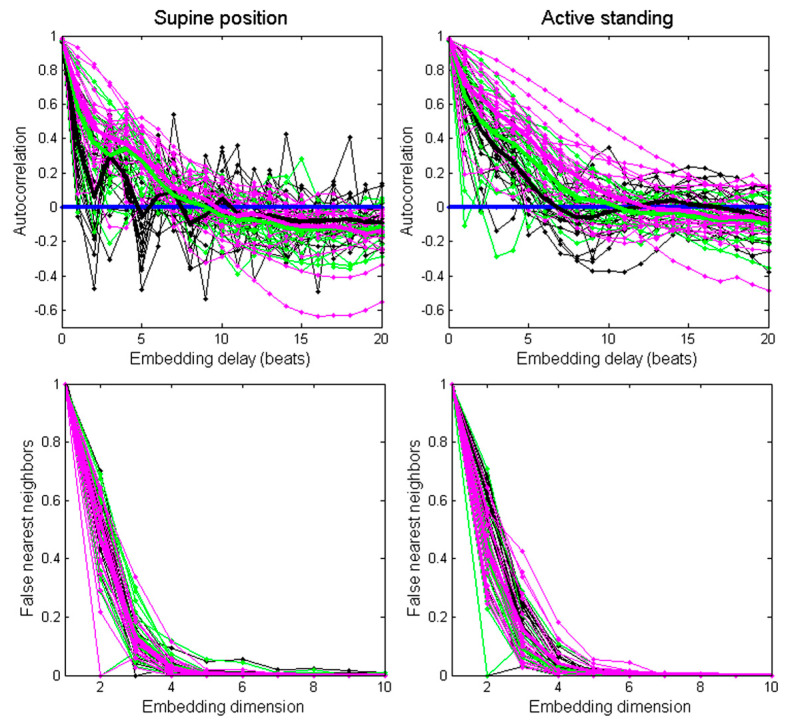
Assessment of the autocorrelation function (**upper panels**) and false nearest neighbors method (**lower panels**) for each recording (thin lines) and averaged values per group (thick lines). Black lines correspond to healthy subjects, green lines correspond to ESRD patients before hemodialysis, and magenta lines correspond to ESRD patients after hemodialysis.

**Figure 2 entropy-22-00114-f002:**
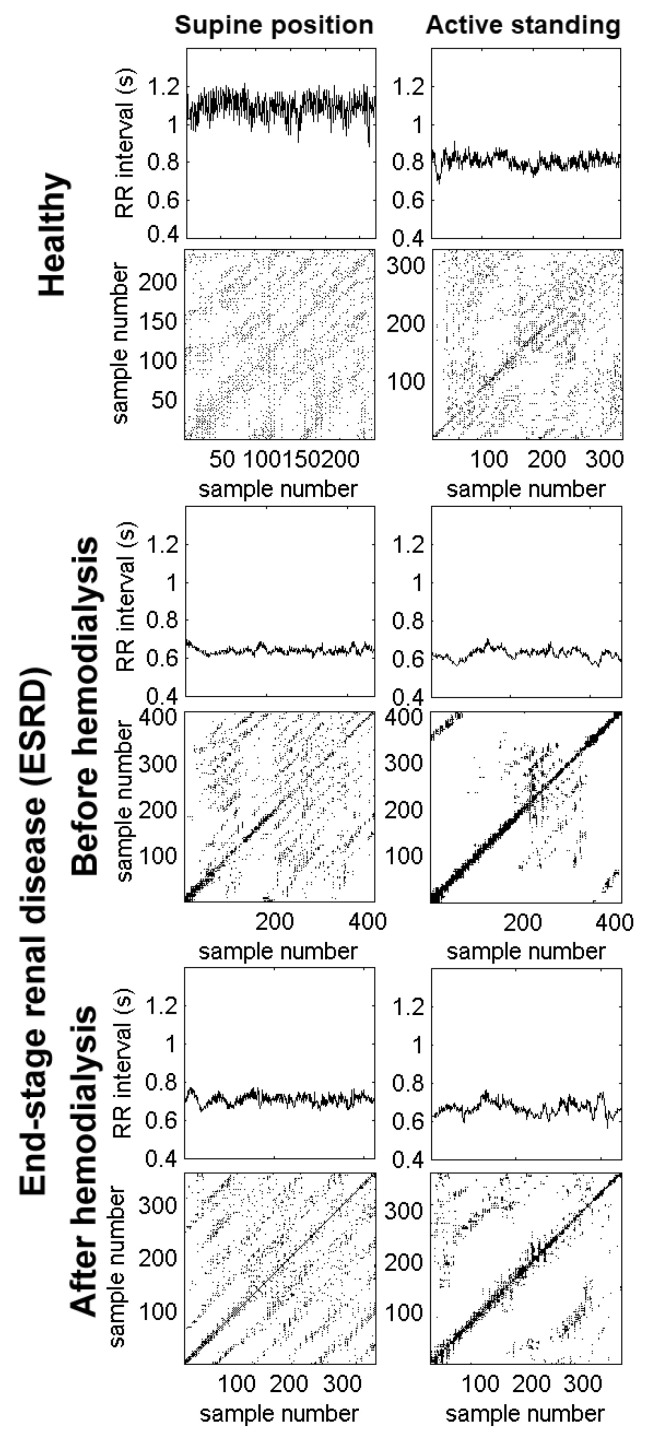
Examples of RR time series and recurrence plots from a healthy subject (**upper panels**) and an end-stage renal disease (ESRD) patient (**lower panels**). The recording from the healthy subject shows a reduction in RR intervals and their variability, as a consequence of changing to an active standing position. The recurrence plot shows extended global diagonal lines formed by short diagonals covering a regularly spaced pattern of the plot surface. In contrast, during the standing position, there is an increase in white zones (i.e., there are less accessible dynamic states in the central area). In the patient before hemodialysis, there is a marked reduction in the RR intervals and their associated variability, even in the supine position. Its recurrence plot quantitative analysis (RQA) plot has less regularly extended diagonals with more concentration around the identity line and its parallel lines. During the active standing condition, the plot is almost empty; there are very fewer accessible dynamical states. In the same patient after hemodialysis, there is a slight improvement in the RR intervals during the supine condition compared to those before hemodialysis, which is also reflected in the RQA plot. Again, during the active standing challenge, there is a reduction in RR intervals and reachable dynamical states.

**Table 1 entropy-22-00114-t001:** Heart rate variability indices evaluated in 30 healthy subjects and 29 end-stage renal disease (ESRD) patients undergoing hemodialysis treatment regularly. Data are shown as mean ± standard deviation. HF: high frequency, LF: low frequency, meanNN: mean cardiac period.

	Healthy	ESRD before Hemodialysis	ESRD after Hemodialysis
Supine position			
Statistical indices			
meanNN (ms)	0.904 ± 0.115 *	0.754 ± 0.106 *^, &^	0.740 ± 0.143 *^, &^
SDNN (ms)	0.058 ± 0.022 *	0.024 ± 0.012 ^&^	0.025 ± 0.011 ^&^
SDSD (ms)	0.051 ± 0.021 *	0.016 ± 0.013 ^&^	0.016 ± 0.012 ^&^
pNN50 (%)	67.2 ± 15.4 *	17.3 ± 19.4 ^&^	20.6 ± 22.8 *^, &^
Spectral indices			
LF (n.u.)	56.7 ± 20.7 *	70.5 ± 20.1 ^&^	66.5 ± 20.0 *
HF (n.u.)	43.8 ± 19.5 *	30.1 ± 19.5 *^, &^	33.7 ± 20.4 *
LF/HF	2.39 ± 3.52 *	4.88 ± 6.62	4.85 ± 7.50
Ln (LF/HF)	0.316 ± 0.990 *	1.025 ± 1.082 *^, &^	0.880 ± 1.138 *
Active standing			
Statistical indices			
meanNN (ms)	0.733 ± 0.112	0.717 ± 0.112	0.616 ± 0.116 ^&, ¶^
SDNN (ms)	0.046 ± 0.021	0.029 ± 0.012 ^&^	0.026 ± 0.022 ^&^
SDSD (ms)	0.025 ± 0.010	0.016 ± 0.013 ^&^	0.013 ± 0.014 ^&^
pNN50 (%)	41.0 ± 17.9	17.8 ± 19.4 ^&^	12.9 ± 19.9 ^&^
Spectral indices			
LF (n.u.)	77.0 ± 15.1	76.4 ± 16.9	77.8 ± 14.7
HF (n.u.)	23.3 ± 15.1	23.6 ± 16.9	23.4 ± 15.5
LF/HF	6.32 ± 8.45	6.65 ± 6.89	6.20 ± 5.48
Ln (LF/HF)	1.391 ± 0.927	1.421 ± 1.030	1.423 ± 0.935

* *p* < 0.05 supine position vs. active standing (same group). ^&^
*p* < 0.05 ESRD group (either before hemodialysis or after hemodialysis) vs. healthy group (same position). ^¶^
*p* < 0.05 before hemodialysis vs. after hemodialysis (same position).

**Table 2 entropy-22-00114-t002:** Recurrence quantification analysis (RQA) indices of heart rate variability evaluated in 30 healthy subjects and 29 end-stage renal disease (ESRD) patients undergoing hemodialysis treatment. Data are shown as mean ± standard deviation. The recurrence plots were reconstructed with the subgroup embedding time delays as described in the Methods section.

	Healthy	ESRD before Hemodialysis	ESRD after Hemodialysis
Supine position			
Determinism	0.384 ± 0.135 *	0.447 ± 0.169	0.446 ± 0.182 *
Mean diagonal length	2.455 ± 0.259	2.535 ± 0.433	2.513 ± 0.395 *
Longest diagonal line	16.884 ± 9.148	12.420 ± 10.048	14.288 ± 21.322
Entropy	0.790 ± 0.222 *	0.872 ± 0.338	0.864 ± 0.351 *
Laminarity	0.410 ± 0.199 *	0.553 ± 0.162 ^&^	0.551 ± 0.182 *^,&^
Trapping time	2.308 ± 0.234 *	2.697 ± 0.486 *^, &^	2.709 ± 0.524 *^, &^
Longest vertical line	6.868 ± 2.714 *	9.354 ± 2.913 ^&^	9.514 ± 3.479 *^, &^
Recurrence time type 1	14.120 ± 1.500	15.734 ± 1.571 ^&^	15.975 ± 1.985 ^&^
Recurrence time type 2	19.631 ± 4.885 *	26.856 ± 7.351 *^, &^	27.683 ± 9.075 *^, &^
Active standing			
Determinism	0.541 ± 0.159	0.506 ± 0.197	0.530 ± 0.193
Mean diagonal length	2.618 ± 0.316	2.661 ± 0.552	2.843 ± 0.736
Longest diagonal line	19.669 ± 11.931	11.626 ± 10.425 ^&^	14.144 ± 11.671
Entropy	0.991 ± 0.284	0.980 ± 0.437	1.128 ± 0.437
Laminarity	0.650 ± 0.147	0.610 ± 0.183	0.629 ± 0.170
Trapping time	2.827 ± 0.391	2.921 ± 0.618 ^&^	3.157 ± 0.696 ^&, ¶^
Longest vertical line	10.135 ± 2.822	9.810 ± 2.773	12.208 ± 3.850 ^&, ¶^
Recurrence time type 1	14.985 ± 1.256	15.555 ± 1.769	15.322 ± 2.677
Recurrence time type 2	28.644 ± 7.399	30.858 ± 10.643	31.259 ± 10.295

* *p* < 0.05 supine position vs. active standing (same group). ^&^
*p* < 0.05 ESRD group (either before hemodialysis or after hemodialysis) vs. healthy group (same position). ^¶^
*p* < 0.05 before hemodialysis vs. after hemodialysis (same position).

**Table 3 entropy-22-00114-t003:** Pearson’s correlation coefficients between meanNN and linear indices of heart rate variability in short-term electrocardiogram (ECG) recordings. The estimation of each correlation coefficient (r) includes data obtained at both the supine position and during active standing. LFnu: negative correlation, HFnu: positive correlation.

	Healthy		ESRD before Hemodialysis		ESRD after Hemodialysis	
	r	*p*-Value	R	*p*-Value	R	*p*-Value
Statistical indices						
SDNN	0.545	<0.001	0.475	<0.001	0.453	<0.001
SDSD	0.764	<0.001	0.448	<0.001	0.665	<0.001
pNN50	0.672	<0.001	0.524	<0.001	0.768	<0.001
Spectral indices						
LFnu	−0.366	0.004	−0.223	0.093	−0.627	<0.001
HFnu	0.385	0.002	0.245	0.063	0.645	<0.001
LF/HF	−0.122	0.351	−0.232	0.079	−0.43	0.001
Ln (LF/HF)	−0.339	0.008	−0.269	0.041	−0.609	<0.001

**Table 4 entropy-22-00114-t004:** Pearson’s correlation coefficients between meanNN and recurrence quantitative analysis (RQA) indices of short-term ECG recordings. The estimation of each correlation coefficient includes data obtained at both the supine position and during active standing. The recurrence plots were reconstructed with the subgroup embedding time delays, as described in the Methods section.

	Healthy		ESRD before Hemodialysis		ESRD after Hemodialysis	
	R	*p*-Value	R	*p*-Value	R	*p*-Value
Determinism	−0.452	<0.001	−0.256	0.052	−0.485	<0.001
Mean diagonal length	−0.387	0.002	−0.272	0.039	−0.478	<0.001
Longest diagonal line	−0.416	0.001	−0.286	0.029	−0.412	0.001
Entropy	−0.489	<0.001	−0.342	0.009	−0.619	<0.001
Laminarity	−0.485	<0.001	−0.220	0.097	−0.477	<0.001
Trapping time	−0.574	<0.001	−0.312	0.017	−0.608	<0.001
Longest vertical line	−0.751	<0.001	−0.466	0.000	−0.836	<0.001
Recurrence time type 1	−0.163	0.214	0.080	0.548	0.230	0.082
Recurrence time type 2	−0.447	<0.001	−0.215	0.106	−0.337	0.010

**Table 5 entropy-22-00114-t005:** Linear stepwise multiple regression analysis with predicted heart rate variability (HRV) indices, and as independent variables, the meanNN, active standing, end-stage renal disease (ESRD), and hemodialysis conditions. All models included the same sample size (N = 176).

Variables	Standardized β	β (C.I._95%_)	P	R^2^
Predicted HRV index: SDNN	0.411
meanNN	0.584	0.088 (0.070–0.106)	< 0.001	
Active standing	0.185	0.008 (0.003–0.013)	0.004	
ESRD condition	−0.213	−0.011 (−0.017–0.005)	0.001	
Hemodialysis	*Excluded variable*	
Predicted HRV index: Determinism	0.288
meanNN	−0.354	−0.419 (−0.588–0.249)	< 0.001	
Active standing	0.217	0.074 (0.027–0.120)	0.002	
ESRD condition	0.181	0.073 (0.020–0.127)	0.008	
Hemodialysis	*Excluded variable*	
Predicted HRV index: Trapping time	0.287
meanNN	−0.392	−2.162 (−2.952–1.372)	< 0.001	
Active standing	0.190	0.301 (0.083–0.519)	0.007	
ESRD condition	0.142	0.301 (0.083–0.519)	0.035	
Hemodialysis	*Excluded variable*	
Predicted HRV index: Recurrence time type 2	0.191
meanNN	−0.264	−12.745 (−20.129–5.362)	0.001	
Active standing	0.219	3.035 (0.998–5.072)	0.004	
ESRD condition	0.146	2.412 (0.077 – 4.746)	0.043	
Hemodialysis	*Excluded variable*

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
