# Peer review of "Association between Mean Heart Rate and Recurrence Quantification Analysis of Heart Rate Variability in End-Stage Renal Disease"

_entropy, 2020, doi:10.3390/e22010114_

Round 1
Reviewer 1 Report
With the exception of HRV and ESRD, avoid use of acronyms in the manuscript.
Though the Task Force (1996) recommendations are cited and used to justify some of the methods, the deviation from their recommendations is not discussed.
It is surprising that Laborde, Mosely & Thayer’s (2017) recommendations are not incorporated or cited.
Can you provide more information about the healthy sample? A number of variables could have an impact on HRV (exercise, medication use, etc.), what other measures were collected?
Throughout, be clear about which group the authors are referring to (i.e., Patients, healthy controls, ESRD patients). It is not always clear.
Were healthy controls also restricted in diet? Were there any restrictions placed on the health controls?
ESRD before and after groups are the same participants, correct? If so, make this clear throughout.
Need to add greater justification for the methods/design of the study. Is does not seem appropriate to compare pre-/post- ESRD patients to a healthy control at one time point, given that HD itself has an impact on the ESRD patients that is not accounted for in the healthy control group?
Similarly, the calculations that are made in the method section need to be introduced and justified in the introduction.
Table 5 and the regression analyses need further clarification. The study does not seem sufficiently powered to use this type of analysis. In each model, HD is listed but excluded. Also, please justify use of heart rate to predict HRV, given the assumptions of regression.
None of these limitations, or any for that matter, are discussed in the discussion section.
Author Response
Comment 1: With the exception of HRV and ESRD, avoid use of acronyms in the manuscript.
Response: In the revised manuscript we avoided the acronym of the clinical word (hemodialysis, HD) and those of the recurrence plot indices: largest diagonal lines (Lmax), longest vertical line (Vmax), type 1 recurrence time (T1) and type 2 recurrence time (T2). The acronyms of the linear HRV indices (mean NN, SDNN, SDSD, pNN50, LFnu, HFnu, LF/HF) were preserved because these are widely accepted in the HRV literature. Also, the acronym for recurrence plot quantitative analysis (RQA) was maintained because, as an essential concept in the manuscript, it is frequently used.
Comment 2: Though the Task Force (1996) recommendations are cited and used to justify some of the methods, the deviation from their recommendations is not discussed.
Response: We verified that our study agrees with the recommendations of the Task Force (1996), its more recent update (Sassi, 2015), and even with most of the recommendations by Laborde, Mosely & Thayer’s (2017). We consider that is unnecessary to describe and justify deeper all the methodological details, since such technical discussion, despite being relevant and interesting for some academic forums, lies beyond the scope of the present work.
Comment 3: It is surprising that Laborde, Mosely & Thayer’s (2017) recommendations are not incorporated or cited.
Response: We thank the reviewer for suggesting such reference, which is now widely cited in the revised manuscript.
Comment 4: Can you provide more information about the healthy sample? A number of variables could have an impact on HRV (exercise, medication use, etc.), what other measures were collected?
Response: None of the healthy subjects were smokers, and they were not treated with any medication (including oral contraceptives in women) or reported history of intense physical training. This is now mentioned on page 3, lines 106 and 107.
Comment 5: Throughout, be clear about which group the authors are referring to (i.e., Patients, healthy controls, ESRD patients). It is not always clear.
Response: We revised the text carefully to ensure that it is clearer now which group we are referring to (i.e. either healthy subjects or ESRD patients).
Comment 6: Were healthy controls also restricted in diet? Were there any restrictions placed on the health controls?
Response: None of the study participants had restrictions in their diet and, even for the case of ESRD patients, the clinical approach in our institution is to remove the excess of water and waste from the blood through an adequate prescription of hemodialysis, which allows adequate restoration of the nutritional status in their diet without restrictions (as it is now mentioned on page 3, line 119). We did not restrict or monitor the food intake before the study, but all study participants were requested to refrain from any intake of coffee or alcohol during the 24 hours prior to the study. This is now mentioned in page 3, line 135.
Comment 7: ESRD before and after groups are the same participants, correct? If so, make this clear throughout.
Response: Indeed, recordings from the same ESRD patients were obtained during baseline and active standing both before and after hemodialysis sessions as it is now clarified (e.g. page 3, lines 131 to 134).
Comment 8: Need to add greater justification for the methods/design of the study. Is does not seem appropriate to compare pre-/post- ESRD patients to a healthy control at one time point, given that HD itself has an impact on the ESRD patients that is not accounted for in the healthy control group?
Response: Indeed, the study design did not consider repeated measures for the healthy group which could provide the measurements at two different time points to compare with the two different time points of the ESRD patients (i.e. the recordings before and after hemodialysis). This is now mentioned in the study limitations (page 12, lines 451 to 453). In the healthy group it is not feasible to perform an intervention equivalent or comparable to hemodialysis. A potential strategy to gather the same number of data points in the healthy group than those of the ESRD patients would be to obtain recordings from the healthy group twice, with a time interval between recordings similar to the hemodialysis duration (approximately 3.6 hours). We may consider such approach in future studies.
Comment 9: Similarly, the calculations that are made in the method section need to be introduced and justified in the introduction.
Response: We did not address this comment in the revised manuscript as it was not clear for us which calculations the reviewer was referring to. All relevant methodological details are endorsed by proper reference (within the methods section), and we consider that the inclusion of a detailed description and justification of the methodological considerations (either about the HRV analysis or the study design) within the Introduction section would just distract the readers beyond the scope of this work.
Comment 10: Table 5 and the regression analyses need further clarification. The study does not seem sufficiently powered to use this type of analysis. In each model, HD is listed but excluded. Also, please justify use of heart rate to predict HRV, given the assumptions of regression.
Response: All the regression analyses were performed with the same sample size (N = 176), including all recordings from the healthy subjects (during supine position and active standing) and from the ESRD patients (during supine position and active standing, both before and after hemodialysis). Given this sample size, the statistical power is very good even for a moderate size effect. The attached document shows the post hoc calculation of the statistical power as a function of such sample size for a moderate size effect (f2 = 0.2360) calculated from the model with lowest multivariate determinant coefficient (R2 = 0.191, as obtained from Table 5 of our manuscript). The sample size used for the regression analyses is now clarified in Table 5 and in the Statistical Analysis section (page 5, line 221 and 222).
Regarding the use of heart rate to predict HRV, it is now clarified in page 5, line 215 that the RQA indices were considered as the dependent variables (or the outcome variables), and that the meanNN was one of the independent variables (or predictor variables). The use of the mean heart rate to predict HRV is actually the central point of this work, as it is described in the last two paragraphs of the Introduction section.
Comment 11: None of these limitations, or any for that matter, are discussed in the discussion section.
Response: The revised manuscript includes a paragraph in the Discussion Section with relevant limitations of the study (page 12, lines 446 to 453).

Reviewer 2 Report
The study has been submitted to the journal "Entropy". It could therefore be helpful and interesting for the readers who are not experts in Heart Rate Variability to know that the study of Time Series Analysis have been applied with some success by the socalled Maximum Entropy Method, especially taking into consideration that many of the readers do not know the field of cardiology but maybe experts in fields as chaos or entropy and could be interested in Heart Rate Variability after having read the paper.
Author Response
Comment: The study has been submitted to the journal "Entropy". It could therefore be helpful and interesting for the readers who are not experts in Heart Rate Variability to know that the study of Time Series Analysis have been applied with some success by the socalled Maximum Entropy Method, especially taking into consideration that many of the readers do not know the field of cardiology but maybe experts in fields as chaos or entropy and could be interested in Heart Rate Variability after having read the paper.
Response: We thank the reviewer for this suggestion. The revised manuscript now includes a brief discussion (page 12, paragraph 2) about several methods that assess the concept of entropy on heart rate variability time series (including the maximum entropy method). It is very interesting that despite a clear interest for applying the concept of entropy in time series analysis (particularly for heart rate variability), a comprehensive comparison of these methods and their application has not been reported. It would be interesting to see such important comparison published in the future. Moreover, the correlation between the entropy indices (for any of those reported methods) and the mean heart rate has not been assessed, and given our current finding that the Shannon’s entropy (based on the recurrence plot analysis) showed a significant negative correlation with the meanNN, it would be justified to further investigate the interpretation of the entropy estimated with other methods in relation to its dependence on mean heart rate.
Round 2
Reviewer 1 Report
The authors have adequately addressed my concerns in their revision.